# LGR5 Expression Predicting Poor Prognosis Is Negatively Correlated with WNT5A in Colon Cancer

**DOI:** 10.3390/cells12222658

**Published:** 2023-11-20

**Authors:** Lubna M. Mehdawi, Souvik Ghatak, Payel Chakraborty, Anita Sjölander, Tommy Andersson

**Affiliations:** Cell and Experimental Pathology, Department of Translational Medicine, Lund University, Skåne University Hospital, SE 214 28 Malmö, Sweden; souvik.ghatak@med.lu.se (S.G.); payel.chakraborty@med.lu.se (P.C.); anita.sjolander@med.lu.se (A.S.)

**Keywords:** WNT5A, Foxy5, LGR5, R-spondin3, WNT/β-catenin signaling, VEGFA

## Abstract

WNT/β-catenin signaling is essential for colon cancer development and progression. WNT5A (ligand of non-canonical WNT signaling) and its mimicking peptide Foxy5 impair β-catenin signaling in colon cancer cells via unknown mechanisms. Therefore, we investigated whether and how WNT5A signaling affects two promoters of β-catenin signaling: the LGR5 receptor and its ligand RSPO3, as well as β-catenin activity and its target gene *VEGFA*. Protein and gene expression in colon cancer cohorts were analyzed by immunohistochemistry and qRT-PCR, respectively. Three colon cancer cell lines were used for in vitro and one cell line for in vivo experiments and results were analyzed by Western blotting, RT-PCR, clonogenic and sphere formation assays, immunofluorescence, and immunohistochemistry. Expression of WNT5A (a tumor suppressor) negatively correlated with that of LGR5/RSPO3 (tumor promoters) in colon cancer cohorts. Experimentally, WNT5A signaling suppressed β-catenin activity, LGR5, RSPO3, and VEGFA expression, and colony and spheroid formations. Since β-catenin signaling promotes colon cancer stemness, we explored how WNT5A expression is related to that of the cancer stem cell marker DCLK1. DCLK1 expression was negatively correlated with WNT5A expression in colon cancer cohorts and was experimentally reduced by WNT5A signaling. Thus, WNT5A and Foxy5 decrease LGR5/RSPO3 expression and β-catenin activity. This inhibits stemness and VEGFA expression, suggesting novel treatment strategies for the drug candidate Foxy5 in the handling of colon cancer patients.

## 1. Introduction

WNT signaling is conventionally divided into two main pathways: β-catenin-dependent (canonical) and β-catenin-independent (noncanonical) signaling [1,2]. The dominant pathway is dependent on the specific WNT ligand that triggers the intracellular signal, the receptor(s) it binds to, and the cellular context. There are several ligands, receptors and coreceptors that participate in triggering these signaling pathways. To date [1,2], 19 WNT glycoproteins have been discovered and shown to initiate downstream intracellular signaling by binding distinct cell surface receptors, predominantly one or several of 10 different seven-transmembrane Frizzled (FZD) receptors, in the absence or presence of multiple coreceptors, such as low-density lipoprotein receptor-related proteins (LRP5/6). In addition, WNT proteins have been shown to bind to the tyrosine kinase receptors ROR1, ROR2, RYK and PTK7 [1,3]. It has been found that different WNT proteins have distinct abilities to predominantly activate the β-catenin-dependent or the β-catenin-independent signaling pathway. This has led to WNT3A being commonly used as an activator of β-catenin-dependent signaling and WNT5A often used as an activator of β-catenin-independent signaling in colon cancer cells [4,5].

Immunohistochemical evaluations of primary tumor tissues from breast, colon, hepatocellular and prostate cancer patients illustrate that loss of or reduction in WNT5A protein expression correlates with shorter recurrence-free survival and thus poor prognosis [1,2,6,7,8]. Accordingly, recombinant WNT5A significantly reduces the migration and invasion of breast, colon, hepatocellular and prostate cancer cells [1,2,6,7,8]. However, clinically restoring WNT5A signaling by administering recombinant WNT5A to these patients would not be possible due to the large size and complex posttranslational modifications, but also due to the cell-surface binding property of WNT5A accomplished via its heparan sulfate-binding domain, resulting in its low bioavailability if administered to patients [1,2]. This led to the development of a WNT5A-mimicking hexapeptide, Foxy5, the backbone of which was derived from one of the solvent-exposed amino acid sequences of WNT5A and formylated on its N-terminal methionine [1,2]. Similarly to WNT5A, Foxy5 reduced the migration and invasion of breast and prostate cancer cells [1,2]. Additionally, Foxy5, when administered systemically via intraperitoneal injection in different orthotopic mouse models or injected shortly after cancer cells via tail vein injections, has been shown to impair metastatic spread [1,2]. Moreover, the Foxy5 peptide has been shown in two phase I clinical studies (www.clinicaltrials.gov, accessed on 15 November 2023; NCT02020291 and NCT02655952) to be a safe drug candidate that is well tolerated. The Foxy5 peptide is presently being investigated in a phase II clinical study called Neofox on stage II/III colon cancer patients (www.clinicaltrials.gov, accessed on 15 November 2023; NCT03883802).

In many types of cancer, and in particular in colon cancer, there is a significant increase in WNT-induced β-catenin signaling [9]. Based on the fact that β-catenin signaling is a broad promoter of cancer progression, it is not surprising that significant research has been conducted to find an effective inhibitor of this signaling pathway [9,10]. What is particularly interesting is the fact that WNT/β-catenin-independent signaling has been documented to crosstalk with the WNT/β-catenin-dependent signaling pathway in different contexts [5,11,12,13,14]. We have shown that recombinant WNT5A and/or Foxy5 can significantly inhibit WNT/β-catenin-dependent signaling in colon cancer cells in vitro as well as in vivo [7,15]. However, the mechanism underlying the described crosstalk between WNT5A and Foxy5 and the WNT β-catenin signaling pathway remains unclear.

To address this question, we investigated whether and how WNT5A signaling can interfere with cell surface molecules of colon cancer cells that are known to participate in the regulation of WNT/β-catenin-dependent signaling. The molecule that we focused on was leucine-rich repeat-containing G-protein coupled receptor 5 (LGR5), but we also investigated its ligand, roof plate-specific spondin 3 (RSPO3), since the binding of this ligand to LGR5 has been reported to activate and/or enhance WNT β-catenin signaling [16,17], although an early article reported that it antagonizes WNT signaling and acts as a negative regulator of tumorigenicity [18]. The RSPO family of secreted glycoproteins has four members, and they have been well established as multipotent signaling molecules/ligands in different contexts [17]. Originally in colon cancer, but also later in other cancer types, the *RSPO2* and *RSPO3* gene fusion product has been reported to correlate with a poor prognosis and survival [17]. In addition, *RSPO2* and *RSPO3* mRNAs are significantly overexpressed in a subset of colon cancer [19]. Crucially for the present study, experimental work has revealed that increased RSPO3 activity in the intestine serves as an oncogenic promoter [17].

In this study, we explored a possible relationship between the expression of WNT5A, a favorable prognostic factor in colon cancer, and the β-catenin signaling modulators LGR5 and RSPO3, which are also negative prognostic factors in colon cancer. Potential correlations between WNT5A signaling and LGR5 and RSPO3 were also investigated.

The results from this study could further broaden our understanding of WNT5A signaling and its mechanisms of action and thereby influence how it can be therapeutically targeted in future clinical trials.

## 2. Materials and Methods

### 2.1. Tissue Microarray (TMA) and Immunohistochemistry (IHC)

We utilized a TMA that contained primary tumor samples from patients who underwent colon cancer surgery at Malmö University Hospital (Malmö, Sweden) in 1990. The clinicopathological characteristics of this Malmö colon cancer cohort (Malmö-CC cohort) are outlined in Appendix A. As indicated in this table, we found in a univariate analysis that sex and tumor–node–metastasis (TNM) stage were prognostic factors. All details regarding the study design, patient follow-up, preparation of TMA and IHC protocol were described in our previous study [20]. Sixty-six patient tumor samples were evaluated for the expression of LGR5 using an anti-LGR5 antibody (OTI2A2, Thermo Fisher Scientific, Waltham, MA, USA). Sixty-one patient tumor samples were available for the evaluation of DCLK1 expression, and these samples were analyzed with an anti-DCLK1 (DCAMKL1 D-3) antibody (sc-514684, Santa Cruz Biotechnology, Santa Cruz, CA, USA). Fifty-seven patient tumor samples were available for the evaluation of WNT5A expression, and these samples were analyzed with the anti-WNT5A antibody (AF645, R&D Systems, Inc. Minneapolis, MN, USA). Tumor samples from seventy patients were available for the evaluation of RSPO3 expression, and these samples were analyzed with the anti-RSPO3 antibody (ab233113, Abcam plc, Cambridge, UK). The staining intensity was evaluated by using the immunoreactive score (IRS), where IRS = staining intensity X percentage of positive cells. The intensity staining was scored as follows: 0 = negative, 1 = weak, 2 = moderate and 3 = strong staining. The percentage of positive cells was defined as follows: 1 = 0–25%, 2 = 26–50%, 3 = 51–75% and 4 ≥76%. Each sample thus had an IRS score between 0–12.

### 2.2. Cell Lines

The human colon cancer cell lines HCT116 (ATCC CCL-247^TM^), SW480 (ATCC CCL-288^TM^) and HT-29 (ACTT HTB-38^TM^) were purchased from the American Type Culture Collection (ATCC, Manassas, VA, USA). The cells were cultured following the supplier’s instructions and grown in the presence of 10% fetal bovine serum (FBS), antibiotics and glutamine.

### 2.3. Xenograft Colon Cancer Tissues

We previously injected human HT-29 colon cancer cells subcutaneously into the flanks of female nude mice (BALB/c nu/nu; 5–6 weeks old), and once tumors were established, we treated them intraperitoneally with either vehicle or Foxy5 (2 µg/g) every second day for 14 days [15]. In the present study, we used these xenograft tissues for our analysis of LGR5, RSPO3 and VEGFA mRNA and protein expression.

### 2.4. Real-Time qPCR

HT-29 and HCT116 cells were incubated in the absence or presence of 400 ng/mL recombinant WNT5A (rWNT5A) or 100 µM Foxy5 for 24 h. Thereafter, the cells were washed in phosphate-buffered saline (PBS) after this 24 h incubation period. RNA from cells and xenograft tissue samples was isolated using the Qiagen RNeasy Plus Mini Kit. Cells scraped from the plates or parts of the xenograft tissues were homogenized in the lysis buffer provided in the RNA isolation kit, purified on RNeasy MinElute Spin Columns according to the manufacturer’s instructions and eluted with RNase-free water. cDNA synthesis was performed using RevertAid H Minus M-MuLV reverse transcriptase (Thermo Scientific, Waltham, MA, USA). The following TaqMan probes (from Applied Biosystems, Cambridge, UK) were used: *DCLK1* (Hs00178027-m1), *LGR5* (Hs00173664_m1), *VEGFA* (Hs05484830_s1), *RSPO3* (Hs06473829_m1), and *HPRT1* (Hs99999909_m1). Amplifications were performed with the Mx3005P system (Agilent Technologies Inc., Santa Clara, CA, USA), and the results were normalized against the housekeeping gene *HPRT1* and analyzed with MxPro software 1.10.51.

### 2.5. Clonogenic Assay

Approximately 500 HCT-116 cells/well were seeded into a six-well plate and incubated for 24 h in the absence or presence of either 400 ng/mL rWNT5A or 100 µM Foxy5 for 24 h. Thereafter, the medium was changed to fresh complete medium, and the cells were incubated for six days to allow colony formation to take place. The colonies were then fixed with 4% paraformaldehyde (PFA) and stained with crystal violet. The stained colonies were counted using ImageJ software Version 1.53t (NIH, Bethesda, MD, USA).

### 2.6. Colonosphere Formation

Colonospheres derived from HCT-116 cells were formed according to a protocol described previously [20]. After trypsinization, the cells were counted, and approximately 1000 cells/well were seeded into ultralow attachment round-bottom plates (7007; Corning Inc., Corning, NY, USA). Colonospheres were allowed to form for one week in DMEM-F12 medium supplemented with L-glutamine and antibiotics. The colonospheres were then treated with vehicle (0.09% NaCl) or 100 µM Foxy5 for 48 h, after which proteins were extracted from each well using RIPA lysis buffer.

### 2.7. Western Blotting

Extracted proteins were quantified and used for Western blot analysis using a previously described protocol [7]. The following specific primary antibodies were used: anti-LGR5 antibody (OTI2A2, Thermo Fisher Scientific, Waltham, MA, USA), anti-DCLK1 antibody (DCAMKL1 D-3, sc-514684, Santa Cruz Biotechnology, Santa Cruz, CA, USA), anti-VEGFA antibody (JH121, Thermo Fisher Scientific, Waltham, MA, USA), anti-β-actin antibody-C4 (SC-47778, Santa Cruz Biotechnology, Santa Cruz, CA, USA), and anti-GAPDH -FL-335 (sc-2778, Santa Cruz Biotechnology, Santa Cruz, CA, USA). The secondary antibodies used for Western blotting were peroxidase-linked goat anti-rabbit (p0448, diluted 1:2000) or peroxidase-linked rabbit anti-mouse (P0161, diluted 1:2000, Dako, Glostrup, Denmark). The expression of the targeted proteins was visualized and analyzed with a ChemiDoc XRS+ System (Bio-Rad Laboratories, Hercules, CA, USA).

### 2.8. Immunofluorescence Analysis

Immunofluorescence analysis of SW480 cells was performed as previously described [20]. Briefly, SW480 cells were cultured on coverslips for 24 h and then starved for 1 h prior to the experiments. The cells were then incubated in the absence or presence of 100 µM Foxy5 for 24 h. After this incubation, the cells were washed with PBS, fixed with 4% PFA for 10 min, washed twice with 1x PBS containing 0.05% Tween-20 (PBS-T20) and then permeabilized with 0.01% Triton-X100 for 15 min at room temperature. Cells were washed twice with PBS-T20, blocked with 5% BSA for at least 1 h and then washed 3 times with PBS-T20 (0.1%). Thereafter, the cells were incubated with a cocktail of two primary antibodies, anti-active β-catenin antibody (70034, Cell Signaling Technology, Danvers, MA, USA) and anti-LGR5 antibody (OTI2A2, Thermo Fisher Scientific, Waltham, MA, USA) overnight at 4 °C. After this incubation, the cells were washed 5 times in PBS-T20 (0.1%), followed by incubation with species-specific Alexa Fluor 488- or 546-conjugated antibodies for 1 h before the final DAPI staining for 10 min. The coverslips were washed and mounted on glass slides. Fluorescence images were captured with a Zeiss LSM 700 confocal microscope (Carl Zeiss Microscopy GmbH, Jena, Germany), and the mean fluorescence intensity (MFI) was analyzed for each image using ImageJ region of interest (ROI) algorithm software Version 1.53t (NIH, Bethesda, MD, USA). Graphs were plotted with GraphPad Prism 9.0 (La Jolla, CA, USA).

### 2.9. Acquisition of Gene Expression and Clinical Data from The Cancer Genome Atlas (TCGA) and Gene Expression Omnibus (GEO) Datasets

The normalized RNA-sequencing dataset (TPM) and clinical information associated with colon adenocarcinoma (COAD) samples were downloaded from The Cancer Genome Atlas (TCGA) dataset (https://portal.gdc.cancer.gov/; https://tcpaportal.org/tcpa/; as of 25 August 2022). Out of 320 colon cancer cases, 46 cases with metastasis (stage IV) were eliminated. Thus, 274 stage I, II or III colon cancer cases with clinical information were included in the present study. For further analysis, normalized gene expression data from the TCGA-COAD dataset were log2-transformed. TCGA-COAD data were used to validate the clinical cohort prognostic assessments for *WNT5A*, *LGR5*, *VEGFA* and *DCLK1* (detecting both the *DCLK1A* and the *DCLK1B* isoforms). mRNA microarray data were collected from the GSE44076 dataset for the correlation analysis between genes of interest. The mRNA expression profiles were normalized using the robust multiarray average (RMA) algorithm in R. We downloaded preprocessed data from GEO using the Bioconductor package “GEOquery”.

### 2.10. Identification of Independent Prognostic Parameters of Colon Cancer

To identify and validate the independent prognostic value of WNT5A, LGR5, DCLK1 and VEGFA at the protein and mRNA levels, univariate and multivariate Cox regression analyses were performed in the Malmö-CC and TCGA-COAD cohorts. Parameters with *p* < 0.05 based on the univariate analysis in Appendix A were included in the multivariate Cox regression analysis (adjusted for sex, TNM stage and lymph node metastasis). The patients were divided into high- and low-risk groups according to the optimal cutoffs for WNT5A, LGR5, DCLK1 and VEGFA IRS and mRNA expression level as determined by the receiver-operating characteristic (ROC)–Youden index association criteria.

### 2.11. Statistical Analysis

Statistical analyses were performed using IBM SPSS version 20 (Chicago, IL, USA), MedCalc version 18 (Ostend, Belgium), GraphPad Prism version 9.0 (La Jolla, CA, USA), and R 3.2.4 statistical packages. Statistical differences between protein expression, mRNA expression and various clinicopathologic factors were determined by the chi-squared (χ^2^) test. The Benjamini–Hochberg method was used to correct for multiple hypothesis testing wherever applicable. All statistical tests were two-sided, and a *p* value of <0.05 was considered to indicate statistical significance. OS was defined as the time from the day of surgery to death or the end of follow-up and was analyzed by the log-rank test. We performed ROC curve analysis to evaluate the predictive power of the expression levels of the selected genes. All protein and gene expression values from the Malmö-CC and TCGA-COAD cohorts were used to build an overall survival classifier (OSC) using Cox proportional hazard regression. The Cox proportional hazard (CoxPH) probability values were calculated for each gene according to gene and protein expression levels, and the risk score for overall survival was calculated. To evaluate the association of protein expression in colon cancer tissue with overall survival, univariate and multivariate Cox proportional hazard regression models were applied, and hazard ratios (HRs) together with 95% confidence intervals (CIs) were calculated to determine the risk of death. The multivariate analysis was adjusted for the identified prognostic factors: sex, lymph node metastasis (LNM) and TNM stage (Appendix A). All patients with incomplete or missing information (for the clinical cohort) and missing expression data (for the in silico cohort) were excluded from the analysis. To plot the Kaplan–Meier curves, we dichotomized the patients into low- or high-risk groups based on Youden index-derived cutoff values (X-tile software 3.6.1, Yale School of Medicine). All comparisons between the mean values were performed using Student’s *t* test, and categorical variables were compared with the χ^2^ test. The results are expressed as the mean ± standard error of the mean (SEM). The correlations between the levels of two protein or mRNAs were further analyzed by the nonparametric Spearman correlation method. We performed ROC curve analysis to evaluate the accuracy of the values of each protein and gene in predicting overall survival.

## 3. Results

### 3.1. WNT5A and LGR5 Protein Expression in Colon Cancer Tissue

Colon cancer tissue samples with matched normal mucosa samples from the patients were evaluated by immunohistochemistry (IHC). The clinicopathological characteristics of the included patients are described in Appendix A. In comparison with a previous analysis of WNT5A protein expression in this cohort [7], we optimized the evaluation of the staining to not only consider the staining intensity but also the percentage of positively stained cancer cells, referred to as the immunoreactive score (IRS; see Section 2 Materials and Methods). A comparison of WNT5A and LGR5 protein expression in morphologically normal and cancer tissue patients revealed that normal tissue has a significantly higher WNT5A IRS than tumor tissue (Figure 1A). The calculated IRS difference between normal tissue and colon cancer tissue was 4.09 ± 0.75 (*p* < 0.001). In contrast, LGR5 had a significantly lower IRS in normal adjacent tissue than in tumor tissue (Figure 1B), with a calculated IRS difference of 1.33 ± 0.36 (*p* < 0.0004).

Next, we investigated how WNT5A and LGR5 protein expression levels in colon cancer tissue were related to the overall survival of patients. We have previously shown by a univariate (Kaplan-Meier) overall survival analysis that patients with low WNT5A expression in their colon cancer tissue had a poor prognosis, with a high risk of progression and relapse [7]. Here, we divided the patients into low and high WNT5A expression groups (Figure 1C): we defined low expression of WNT5A as an IRS < 6.5 and high expression as an IRS ≥ 6.5. Using this optimized IRS approach and adjusting for clinically relevant factors (sex of the patients, TNM stage, lymph node metastasis; Appendix A), we performed a multivariate analysis of low-risk (high WNT5A expression) and high-risk (low WNT5A expression) patients (Figure 1D). Multivariate analysis is considered to give a more realistic picture of the actual situation, since it considers the statistically significant clinical factors identified in the univariate analysis (Appendix A). In this multivariate analysis, we found that patients in the low-risk group had significantly longer overall survival than patients in the high-risk group (Figure 1D). The prognostic model based on WNT5A achieved an AUC value of 0.72 in the ROC curve analysis (Appendix A).

It has previously been demonstrated by at least two independent meta-analyses that in contrast to the prognostic effect of WNT5A expression, patients with elevated expression of the LGR5 protein in their colorectal cancer tissue had poor overall survival. However, due to conflicting results, possibly explained by the heterogeneity of LGR5 expression, this topic is still a matter of debate, as outlined in the review article by Morgan et al. [21]. We divided the patient samples in the Malmö-CC cohort into high and low LGR5 expression groups (Figure 1E); we defined low expression of LGR5 as an IRS < 1.5 and high expression as an IRS ≥ 1.5. Using the optimized IRS approach and adjusting for the same clinically relevant factors as above (Appendix A), we performed a multivariate analysis based on the risk score, where “low risk” refers to low LGR5 expression and “high risk” refers to high LGR5 expression. In this multivariate analysis, we found that patients in the low-risk group had significantly longer overall survival than patients in the high-risk group (Figure 1F). The multivariate model had an AUC value of 0.69 in the ROC curve analysis (Appendix A).

### 3.2. Evaluation of Survival Probabilities and Correlation between WNT5A and LGR5 Expression Based on Data from Publicly Available Colon Cancer Cohorts

To validate our WNT5A and LGR5 survival results (Figure 1), we assessed data for 274 patients from the colon cancer cohort TCGA-COAD who provided colon cancer tissue mRNA samples. Although the expression of the WNT5A protein in hepatocellular carcinoma and breast cancer has been shown to be regulated at the translational level [22,23], the reduction in WNT5A protein expression in colon cancer tissue has been shown to be due to methylation of the WNT5A promoter [24]. The data from this public database confirm the overall survival findings showing that patients with high WNT5A mRNA expression in their colon cancer tissues have a significantly better five-year overall survival rate than patients with low WNT5A mRNA expression in their colon cancer tissues (Figure 2A). Likewise, we confirmed that patients with low LGR5 mRNA expression in their colon cancer tissues had a better five-year overall survival rate than patients with high LGR5 mRNA expression in their cancer tissues (Figure 2B). The AUC values for WNT5A and LGR5 mRNA expression in the TCGA-COAD dataset were higher than the AUC values for WNT5A and LGR5 protein expression in the Malmö-CC cohort (Appendix A).

Next, we investigated the predictive value of combining the mRNA expression levels (low and high) of WNT5A and LGR5 in colon cancer tissues and then correlated them with the overall survival of the patients. We separated patients into four groups based on WNT5A and LGR5 expression (high or low for each gene). Unfortunately, the number of patients included in the Malmö-CC cohort (analyzed in Figure 1) was too small to divide the patients into four groups and safely estimate the predictive value of combined expression of WNT5A and LGR5 in terms of the overall survival of the patients. However, this approach of evaluating overall survival based on these four different groups was possible in the larger TCGA-COAD cohort, for which colon cancer tissue samples from 274 patients were available. Our results showed that patients with high WNT5A/low LGR5 expression (Group 1) had a somewhat better prognosis (Figure 2C) than those with high WNT5A expression alone (Figure 2A) or low LGR5 expression alone (Figure 2B). Likewise, patients with low WNT5A/high LGR5 expression (Group 4) had a somewhat worse prognosis (Figure 2C) than those with low WNT5A expression alone (Figure 2A) or high LGR5 expression alone (Figure 2B). These results showed that the difference between Groups 1 and 4 in these analyses is more pronounced and significant (Figure 2C, *p* < 0.0001) than that for the groups based only on WNT5A (Figure 2A) or LGR5 (Figure 2B) expression.

In the GSE44076 colon cancer cohort (N = 198), we next investigated a possible correlation between WNT5A mRNA and LGR5 mRNA expression. We observed a relatively strong negative correlation between WNT5A and LGR5 (Figure 2D). This observation might indicate that the WNT5A ligand regulates the expression of the LGR5 cell surface receptor.

### 3.3. WNT5A Signaling Decreases the Expression of LGR5 and the Growth of Colon Cancer Cells Both In Vitro and In Vivo

To test if WNT5A signaling regulates the expression of LGR5, we investigated how activation of WNT5A signaling, primarily with the WNT5A-mimicking peptide Foxy5, affected LGR5 expression both in vitro and in vivo. To optimize the in vitro conditions for LGR5 expression for such experiments, we set up a 3D model of colonospheres derived from HCT-116 cells (Figure 3A). The HCT-116 cells were grown in an ultralow attachment round-bottom plate for one week before they were stimulated with the WNT5A-mimicking peptide Foxy5 or vehicle control for 48 h. The spheres were first viewed under a microscope, and we observed that the vehicle-treated spheres formed microspheres (Figure 3A). In contrast, Foxy5-treated spheres were smooth at their margins with no microspheres (Figure 3A), suggesting impaired growth and/or spreading of cancer cells. Next, the spheres were washed and lysed, and their LGR5 content was analyzed by Western blotting. The results showed that the expression of LGR5 was significantly reduced by Foxy5 treatment (Figure 3B). We also stimulated HT-29 cells with Foxy5 for 24 h in vitro and found a significant downregulation of LGR5 mRNA in these cells (Figure 3C). To further explore the findings that WNT5A signaling might affect cancer cell behavior, we next examined the effect of WNT5A signaling on the colony formation ability of HCT-116 cells. We performed these clonogenic assays in the absence (vehicle) or presence of recombinant WNT5A (rWNT5A) or Foxy5 for 48 h. We found that, in comparison with vehicle controls, the number of colonies was significantly decreased when the HCT-116 cells were stimulated with either rWNT5A or Foxy5 (Figure 3D). Based on how WNT5A signaling affected LGR5 expression in vitro, we wanted to confirm this effect in an in vivo setup. For this purpose, we used a colon cancer xenograft mouse model in which HT-29 colon cancer cells were inoculated subcutaneously into both flanks of the mice. After tumors were formed, the mice were randomized into two groups, one treated with intraperitoneal injection of vehicle control and the other with Foxy5 every second day from day 7 to day 23. Following termination of the experiment, the tumor tissues were used for isolation of mRNA and fixed for subsequent examination of protein content by IHC. The results revealed that Foxy5-induced signaling led to statistically significant reductions in both LGR5 mRNA expression (Figure 3E) and LGR5 protein expression (Figure 3F). The results regarding WNT5A signaling, LGR5 expression and the growth of colon cancer cells are in line with the role of LGR5 in promoting β-catenin-dependent signaling.

### 3.4. Parallel Analysis of Foxy5-Induced Effects on LGR5 Expression and β-Catenin Signaling

Based on the ability of LGR5 to potentiate β-catenin-dependent signaling, we decided to simultaneously investigate the effect of Foxy5 on both LGR5 expression and β-catenin activity. SW480 colon cancer cells were stimulated with Foxy5 for 24 h followed by double staining with both an anti-LGR5 antibody and an anti-β-catenin antibody. The anti-β-catenin antibody we used only detects unphosphorylated/active β-catenin. The results revealed that Foxy5 induced a decrease in the expression of both LGR5 and active β-catenin (Figure 4A–C). The Foxy5-induced effect on β-catenin activity was further optimized by analyzing only unphosphorylated β-catenin in the nuclei (Figure 4D). We have previously documented, with the same anti-β-catenin antibody, that Foxy5 impairs β-catenin activity in HT-29 cells both in vitro [7] and in vivo [15]. In this context, we also analyzed the expression of CTNNB1 and WNT5A transcripts in the GSE44076 cohort. Despite the fact that this approach does not reflect the activity status of β-catenin, we were able to detect a weak negative correlation between CTNNB1 and WNT5A transcript levels (Figure 4E).

### 3.5. WNT5A Signaling Regulates the Expression of the LGR5 Ligand RSPO3

The four members of the RSPO family of proteins (RSPO1-4) have been shown to stimulate/potentiate WNT/β-catenin signaling, at least in part via their abilities to act as natural ligands for LGR4-6 [17]. The demonstration that RSPO3 is a driver of intestinal cancer [25], and the fact that LGR5 is a WNT/β-catenin signaling target [17] urged us to explore a possible relationship between RSPO3 and LGR5 as well as WNT5A signaling in colon cancer tissues. In the Malmö-CC cohort, IHC analysis revealed significantly elevated protein levels of RSPO3 in colon cancer tissues in comparison with normal colon mucosa (Figure 5A). Based on the elevated levels of both LGR5 and RSPO3 in colon cancer tissues (Figure 1B and Figure 5A), we performed an analysis that revealed positive correlations between RSPO3 and LGR5 at both the protein (Figure 5B) and mRNA (Figure 5C) levels. Next, we investigated a possible correlation between RSPO3 and WNT5A transcripts. We found a negative correlation between these two transcripts in the GSE44076 cohort (Figure 5D). Further, in vitro stimulation of HT-29 cells with Foxy5 for 24 h resulted in a significant downregulation of RSPO3 mRNA in these cells (Figure 5E). To explore this last finding further, we investigated how Foxy5 treatment affected the RSPO3 mRNA and RSPO3 protein levels in colon cancer tissues from our xenograft mouse model. We observed significant reductions in both RSPO3 mRNA and protein expression levels in Foxy5-treated mice in comparison to vehicle (control)-treated mice (Figure 5F,G).

### 3.6. WNT5A Signaling Negatively Correlates with VEGFA Expression in Colon Cancer Tissues

Cancer cells need oxygen and nutrients to proliferate and grow, making neovascularization an essential process for cancer progression. Vascular endothelial growth factor A (VEGFA) is the dominant and most studied angiogenic factor for the progression of solid cancer tissue; therefore, it is logical that it is also secreted from the cancer cells themselves. Importantly for the present study, VEGFA is a downstream β-catenin signaling target. Therefore, we examined the prognosis of colon cancer patients who express high levels of VEGFA. The results from the TCGA-COAD cohort showed that patients expressing high levels of VEGFA mRNA had a poor prognosis (Figure 6A). Patients in the same cohort who had low levels of VEGFA and high levels of WNT5A had a good prognosis (Figure 6B). Furthermore, Spearman’s correlation test between WNT5A and VEGFA, performed in the GSE44076 cohort, revealed a strong significant negative correlation between WNT5A and VEGFA (Figure 6C) but a strong positive correlation between CTNNB1 and VEGFA gene transcript expression in the same GSE44076 cohort (Figure 6D). Based on this finding, we next performed in vitro experiments to investigate whether WNT5A signaling regulates VEGFA expression. We stimulated HT-29 and HCT-116 cells in vitro with rWNT5A or Foxy5 for 24 h and found that both compounds caused clear downregulation of VEGFA mRNA in HT-29 cells (Figure 6E) and VEGFA protein in HCT-116 cells (Figure 6F). To further validate our findings, we investigated how Foxy5 treatment affected VEGFA mRNA and protein levels in colon cancer tissues from a xenograft mouse model. The results revealed a clear reduction in VEGFA mRNA expression in the colon cancer tissue from Foxy5-treated mice when compared to that from vehicle-treated mice (Figure 6G), but the reduction was not statistically significant due to large standard deviations and a limited number of mice. However, the protein expression of VEGFA in these tissue xenografts was significantly downregulated in the Foxy5-treated mice (Figure 6H) compared with the vehicle (0.09% NaCl)-treated control mice. These results confirm that Foxy5 also impairs β-catenin signaling in colon cancer tissue in vivo, since VEGFA is a downstream target of β-catenin signaling. The fact that VEGFA is an essential angiogenesis factor suggests a novel therapeutic use for the drug candidate Foxy5.

### 3.7. Correlations between WNT5A and the Colon Cancer Stem Cell Marker DCLK1 in Colon Cancer Tissue from Different Cohorts

In addition to and most likely due to the abilities of LGR5 and RSPO3 to regulate β-catenin signaling, they are also employed as cancer stem cell markers [16,17,19]. Based on the present observations of how WNT5A signaling relates to and regulates LGR5 and RSPO3 expression levels and our previous observation that Foxy5 downregulates DCLK1 in colon cancer tissues in a xenograft mouse model [15], we decided to characterize DCLK1 expression, and in particular its relation to WNT5A expression in three different colon cancer cohorts. In the smaller Malmö-CC cohort used for IHC analysis of protein expression, we found that colon cancer tissue expressed significantly higher DCLK1 protein levels than matched normal mucosa (Figure 7A). Next, we divided these patient samples into low- and high-expression groups depending on the IRS of DCLK1 (low-expression group if IRS was <6.0 and high-expression group if IRS was ≥6.0) (Figure 7B). Univariate analysis revealed that patients expressing high levels of DCLK1 in their colon cancer tissues had a significantly (*p* < 0.01) worse prognosis than patients with low DCLK1 expression in their cancer tissues (Figure 7C). Even when we adjusted for clinical factors (Appendix A) and performed a multivariate analysis of overall survival, the difference between patients in the low-risk group (low expression of DCLK1) and patients in the high-risk group (high expression of DCLK1) remained statistically significant (Figure 7D). These findings were validated and confirmed by analyzing how DCLK1 mRNA expression was related to overall survival in the larger TCGA-COAD cohort (Figure 7E). We obtained high AUC values for DCLK1 expression in both the Malmö-CC (70%) cohort and the TCGA-COAD (95%) cohort (Appendix A). Next, we investigated the predictive value of combining the mRNA expression levels (low and high) of WNT5A and DCLK1 in colon cancer tissues and then correlated the expression profiles with the overall survival of the patients. This approach results in four groups based on the combination of low or high expression of WNT5A and DCLK1. Our results showed that patients with high WNT5A levels and low DCLK1 levels in their cancer tissue (Group 1) had a somewhat better prognosis (Figure 7F) than those grouped based only on low DCLK1 expression (Figure 7E). However, patients with low WNT5A levels and high DCLK1 levels in their cancer tissue (Group 4) had a significantly worse prognosis (Figure 7F) than those grouped based only on high DCLK1 expression (Figure 7E). This means that the difference between Groups 1 and 4 in these analyses (Figure 7F) is larger than that observed for the low vs. high DCLK1 expression groups (Figure 7E). Based on these results, we next performed an analysis and found a strong negative correlation (R = −0.56) between WNT5A and DCLK1 gene transcript expression in the GSE44076 colon cancer cohort (Figure 7G). These analyses of patient colon cancer tissues provide clinical support for our finding that Foxy5 treatment of mice with colon cancer xenografts caused a reduction in DCLK1 mRNA (Appendix A). These in vivo data were confirmed in vitro by demonstrating that Foxy5 stimulation significantly decreased the expression of the DCLK1 protein in HCT-116-derived colonospheres (Figure 7H). Likewise, Foxy5 stimulation of HT-29 cells in vitro significantly decreased their mRNA expression of DCLK1 (Figure 7I).

We also investigated the combined prognostic value of DCLK1 and LGR5 expression levels. We divided the patients in the TCGA-CODA cohort into two groups: the first group of patients had low expression levels of both DCLK1 and LGR5 in their primary tumors, whereas the other group of patients had high expression levels of both markers. We investigated the overall survival of the patients in these two groups. As expected, the results showed that patients expressing low levels of both DCLK1 and LGR5 had better overall survival than patients with primary tumors with high expression levels of these two stemness markers (HR = 4.04, 95% CI = 1.82–7.05, *p* < 0.0001; Appendix A), with a high AUC value (89%; Appendix A). More interestingly, univariate survival curves revealed that the combined expression profiles of LGR5 and DCLK1 can identify the poor-prognosis group with higher statistical significance and better separation than DCLK1 or LGR5 expression alone. Additionally, the hazard ratio was significantly higher when using a combination of LGR5 and DCLK1 expression.

## 4. Discussion

Despite the development of novel and more advanced targeted treatment strategies, colon cancer remains the third-most common cancer and the second-leading cause of cancer-related death worldwide [26]. A major problem underlying this situation is the lack of treatments that effectively eliminate all cancer cells and/or prevent their dissemination. Therefore, it is vital to find successful treatment strategies to impair the pro-oncogenic signaling events driving recurrence of the disease.

In colon cancer, the WNT5A molecule has been predominantly defined as a cancer suppressor based on findings that patients with low WNT5A expression in their colon cancer tissue had a poor prognosis and the ability of WNT5A signaling to counteract essential cancer-promoting activities [1,2,3,4,24]. One important cancer suppressor function of WNT5A is its ability to impair β-catenin signaling [7,15], a fundamental cancer promoter signal in this cancer type. However, to date, our knowledge regarding how the extracellular ligand WNT5A or its mimicking small peptide Foxy5 inhibits β-catenin signaling is vague. We hypothesized that one plausible mechanism for the inhibition of β-catenin signaling might be its interference with LGR5 expression, since LGR5 has been shown to be an essential part of a complex that potentiates β-catenin signaling in colon cancer cells [21]. To investigate such a possible mechanism, we started by assessing the correlation between WNT5A and LGR5 expression levels in one and the same colon cancer patient cohort.

In the Malmö-CC cohort, we found decreased protein expression of WNT5A in colon cancer tissue in comparison with noncancer colon mucosa, whereas in the same samples, we observed increased LGR5 protein expression. The observation that WNT5A expression in colon cancer is regulated at the transcriptional level [24], in contrast to breast cancer and hepatocellular carcinoma, where it is regulated at the translational level [1,2], made it possible for us to correlate WNT5A and LGR5 expression not only at the protein level but also at the mRNA level in the larger and publicly available TCGA-COAD cohort. In both cohorts, we confirmed that high expression of WNT5A correlated with a good prognosis, whereas high expression of LGR5 correlated with a poor prognosis. If we combined these expression levels, it was clear that high WNT5A and low LGR5 expression was the expression profile correlated with the best prognosis. In the TCGA-COAD cohort, we also observed a relatively strong negative correlation between *WNT5A* and *LGR5* expression (R = 0.37). This suggests that the extracellular ligand WNT5A might regulate the expression of the LGR5 receptor. In a series of in vitro experiments, we demonstrated that WNT5A signaling caused statistically significant downregulation of LGR5 mRNA and protein expression. These findings were paralleled by similar in vivo changes in the expression of the LGR5 ligand RSPO3. These reduced expression levels of LGR5 and RSPO3 occurred in parallel with reduced expression levels of unphosphorylated (active) β-catenin and its downstream target VEGFA in colon cancer cells.

In the present study, we found a strong negative correlation between WNT5A and VEGFA expression in a colon cancer cohort (GSE44076) and that Foxy5 induced a limited (approximately 40%) downregulation of VEGFA expression in colon cancer cells and in colon cancer tissue. This finding reflects a novel mechanism and might contribute to the cancer-suppressive property of the WNT5A-mimicking peptide Foxy5 [1,2]. It is reasonable to assume that a partially reduced expression of VEGFA in a solid cancer will result in a reduction or delay in angiogenesis and thereby a decrease in cancer growth. However, and perhaps even more interesting, is that it might cause a vascular normalization that would make drug delivery more efficient in a combination therapy setting [27].

Solid tumors are composed of many cell types, including inflammatory cells, vascular cells, fibroblasts, and cancer cells. Most of the cancer cells in the tumor have limited capability of self-renewal, except for a minor subpopulation of cancer cells that have the ability of self-renewal and can form a tumor [28]. These cells are referred to as either cancer stem cells or cancer-initiating cells, and the presence of colon cancer stem cells has been shown to correlate with disease relapse [29]. This observation can at least in part be explained by their resistance to chemotherapy and radiotherapy [30].

Together with its ligand RSPO3, the LGR5 receptor promotes β-catenin signaling in cancer cells predominantly via membrane clearance of two endogenous negative regulators of β-catenin signaling: the transmembrane ubiquitin ligases ZNRF3 and RNF43 [17,21,31]. The fact that β-catenin signaling stimulates the colon cancer stem cell niche [32] explains why LGR5 has been recognized and used as a colon cancer stem cell marker [33]. This is in contrast to the stem cell marker DCLK1, which is not a regulator of β-catenin signaling [34,35]. Interestingly, both in colon cancer tissue and in inflammatory intestinal mucosa, a discrepancy between these two frequently used stem cell markers has been reported [36,37]. There are several possible explanations for such discrepancies, including different isoforms of DCLK1 [35], the plasticity of LGR5 expression [38] and the presence of different subpopulations of colon cancer stem cells [39]. To fully evaluate such a possible inconsistency between the cancer stem cell markers LGR5 and DCLK1, we also investigated the expression and prognostic value of DCLK1 in the two colon cancer cohorts and how it correlated with WNT5A expression. In these cohorts, we found a strong prognostic value of DCLK1 expression, similar to that of LGR5. The prognostic value could be further enhanced if these two cancer stem cell markers were combined. As previously found for LGR5, we also found a strong (R = 0.56) negative correlation between DCLK1 and WNT5A expression. These data suggest that DCLK1 expression could be regulated by WNT5A signaling, a notion that was confirmed by our in vitro and in vivo findings that Foxy5 impairs the expression of both DCLK1 mRNA (Figure 7I and Appendix A) and protein (Figure 7 and Ref. [15]). To confirm that LGR5 and DCLK1 are both colon cancer stem cell markers and are related to WNT5A signaling, we studied colony and spheroid formation of colon cancer cells, assays used to demonstrate the presence of cancer stem cells [40]. The fact that both recombinant WNT5A signaling reduced colony and microspheroid formation indicates that WNT5A signaling can reduce the niche of colon cancer stem cells.

The mechanisms underlying these observations are at present hard to delineate, since there might exist a positive feedback loop where a WNT5A signaling-induced reduction in LGR5 and RSPO3 expression leads to reduced β-catenin signaling that will further reduce the expression of LGR5 and RSPO3 [21,25,30], or alternatively, WNT5A signaling causes impaired β-catenin signaling that will reduce the expression of LGR5 and RSPO3. Clearly, additional work on this subject is needed to determine the detailed mechanism(s) responsible for these observations.

## 5. Conclusions

In conclusion, we believe that the present observations point to and strengthen the idea that therapeutic mimicking of WNT5A signaling could—via impaired β-catenin signaling—be an attractive means to reduce the niche of colon cancer stem cells and thereby prolong recurrence-free survival of colon cancer patients.

## Figures and Tables

**Figure 1 cells-12-02658-f001:**
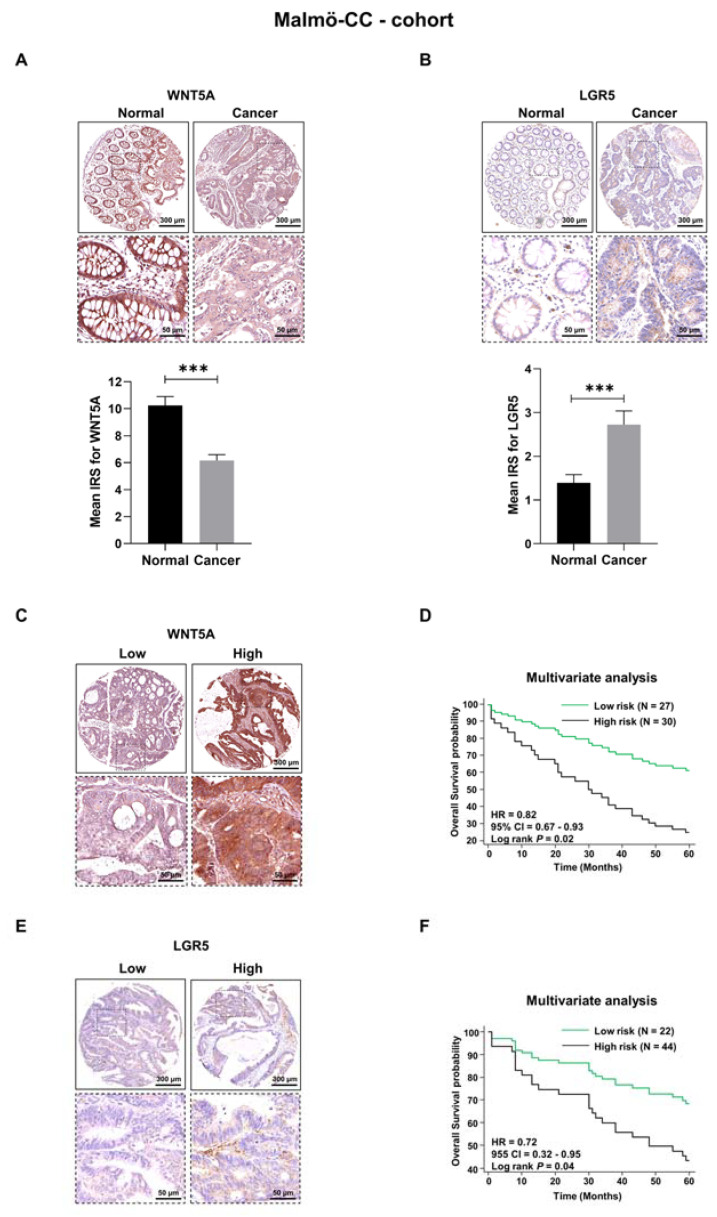
Expression of WNT5A and LGR5 proteins in colon cancer tissue and their relation to the survival of these patients. (**A**) Representative IHC staining images of the WNT5A protein in normal mucosa and matched colon cancer tissue and their mean IRS values are presented in the graph. (**B**) Representative IHC staining images of the LGR5 protein in normal mucosa and matched colon cancer tissue and their mean IRS values are presented in the graph. (**C**) Representative IHC staining images of low and high WNT5A protein expression in colon cancer tissue. (**D**) Multivariate overall five-year survival analysis for low-risk (high WNT5A expression) and high-risk (low WNT5A expression) patients when adjusted for sex, LNM and TNM stage (Appendix A). (**E**) Representative IHC staining images of low and high LGR5 protein expression in colon cancer tissue. (**F**) Multivariate overall five-year survival analysis for low-risk (low LGR5 expression) and high-risk (high LGR5 expression) patients when adjusted for the prognostic factors sex and TNM stage (Appendix A). The results in Panels **A** and **B** are shown as the mean ± SEM; *** *p* < 0.001, analyzed with Wilcoxon’s match paired Student’s *t* test.

**Figure 2 cells-12-02658-f002:**
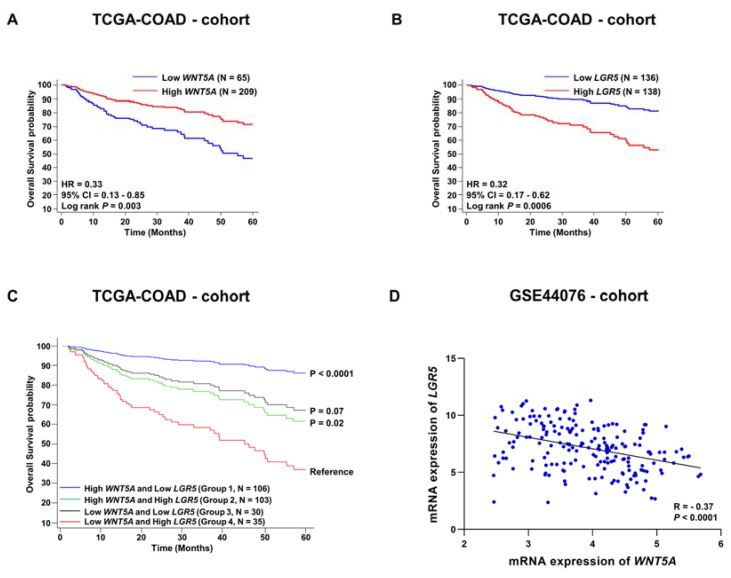
mRNA expression levels of WNT5A and LGR5 in colon cancer tissue and their correlations with the survival of colon cancer patients based on data from publicly available cohorts. Using the TCGA-COAD cohort, we performed univariate five-year overall survival analysis for (**A**) WNT5A and (**B**) LGR5. (**C**) Five-year overall survival analysis of 4 possible groups based on the combinations of WNT5A (low and high) and LGR5 (low and high) mRNA expressions. (**D**) Spearman’s correlation test between WNT5A and LGR5 mRNA expressions using the publicly available GSE44076 cohort.

**Figure 3 cells-12-02658-f003:**
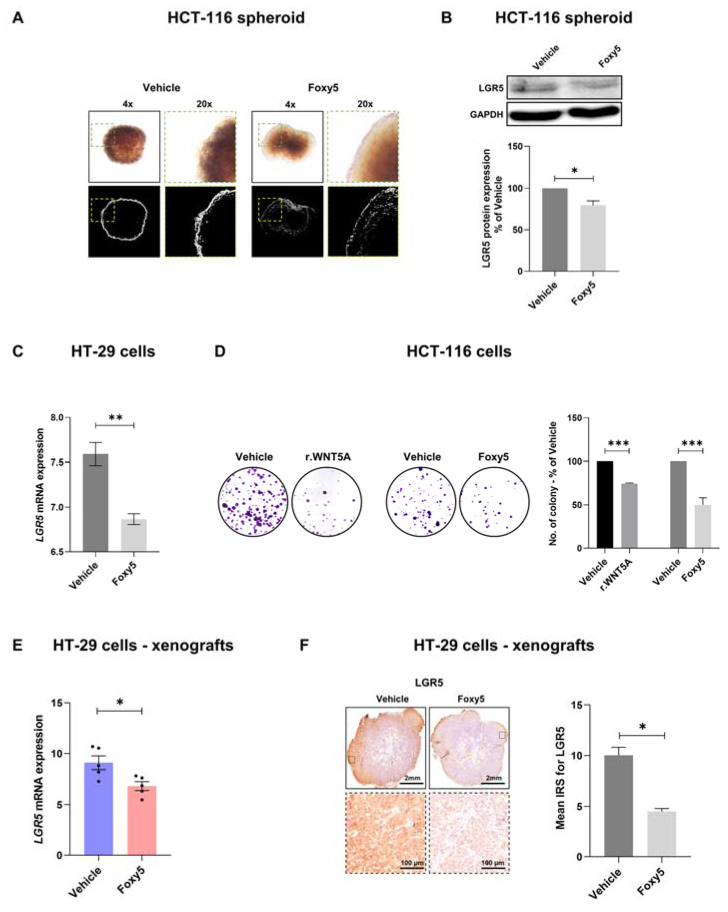
Effects of Foxy5 on colony formation and LGR5 expression in vitro and in vivo. (**A**) HCT-116 colon cancer cells were grown to form colonospheres to optimize the expression of LGR5. Thereafter, the cells were treated with either vehicle or 100 µM Foxy5 for 48 h. The depicted images of the colonospheres were taken after these treatments using inverted phase-contrast microscopy without (upper images) or with ImageJ analysis (lower images). (**B**) The effects of Foxy5 treatment on LGR5 expression in these colonospheres were analyzed by Western blotting and the results are presented in the graph. (**C**) HT-29 colon cancer cells treated in vitro with vehicle or 100 µM Foxy5 for 24 h, after which LGR5 mRNA expression was analyzed. (**D**) Representative images of colony formation of HCT-116 cells stimulated with vehicle (0.1% BSA in water) or with 400 ng/mL rWNT5A and with vehicle (NaCl) or 100 µM Foxy5 for 24 h are shown. The results in panels (**B**–**D**) are presented in the graphs as the mean ± SEM for at least three separate experiments. (**E**) LGR5 mRNA expression in HT-29 colon cancer xenografts was evaluated by qRT-PCR and the results are presented in the graph. (**F**) Representative IHC images of LGR5 protein expression in the HT-29 cell xenografts from mice treated with vehicle or Foxy5 (2 µg/g) every other day for a 14-day period. The results in panels (**E**,**F**) are presented as the mean ± SEM, and 5 xenograft tissues from 5 different mice were analyzed per group of animals. * *p* < 0.05, ** *p* < 0.01 and *** *p* < 0.001, analyzed with Mann-Whitney unpaired Student’s *t* test.

**Figure 4 cells-12-02658-f004:**
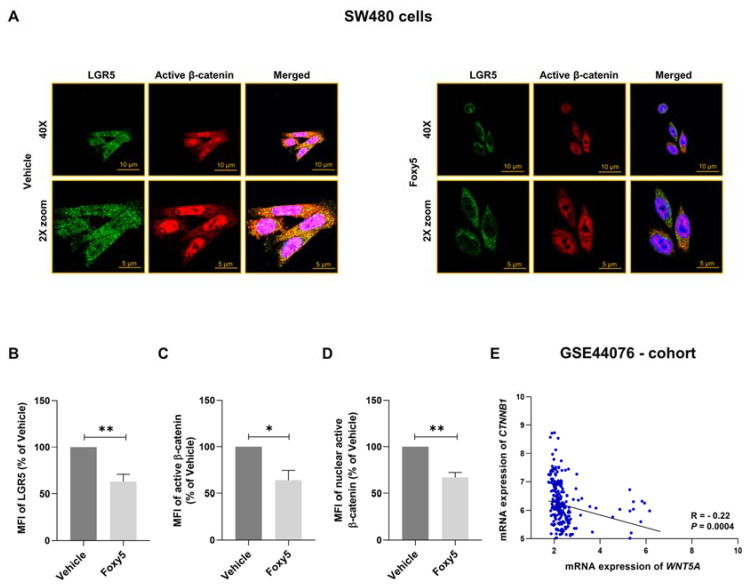
Effects of Foxy5 on LGR5 and β-catenin in colon cancer cells and correlation between β-catenin and WNT5A transcript expressions. SW480 colon cancer cells were treated with either vehicle or 100 µM Foxy5 for 24 h. (**A**) Representative immunofluorescence images of LGR5 and active β-catenin in SW480 cells treated with vehicle (left panels) or Foxy5 (right panels). The scale bars represent 10 µm in the upper panel and 5 µm in the lower panel. The mean fluorescence intensity (MFI) of (**B**) LGR5, (**C**) total active β-catenin, and (**D**) nuclear active β-catenin are outlined in their respective graphs and shown as the mean ± SEM for at least three independent experiments. * *p* < 0.05 and ** *p* < 0.01, analyzed with Mann-Whitney unpaired Student’s *t* test. (**E**) Spearman’s correlation test reveals a correlation between the expression of CTNNB1 and WNT5A transcripts based on data from the GSE44076 cohort.

**Figure 5 cells-12-02658-f005:**
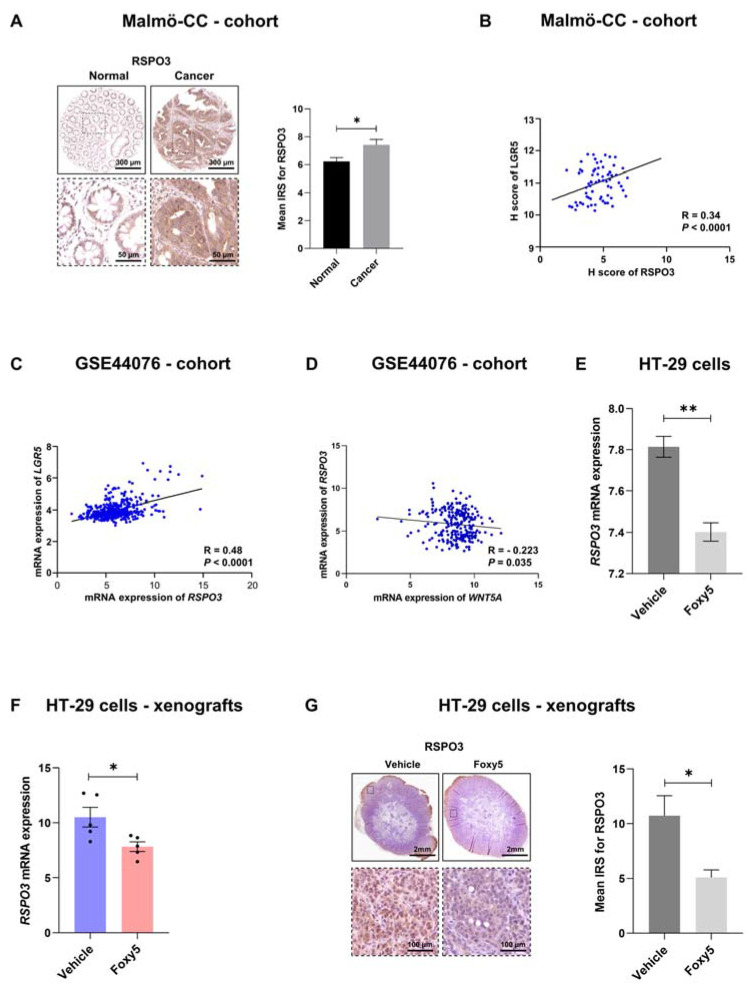
The expression of the LGR5 ligand RSPO3 correlates positively with LGR5 expression and negatively with WNT5A expression in colon cancer tissue and is downregulated by Foxy5. (**A**) Representative IHC staining images of the RSPO3 protein in normal mucosa and matched colon cancer tissue and their mean IRS values are presented in the graph. The results are shown as the mean ± SEM; * *p* < 0.05, analyzed with Wilcoxon match-paired Student’s *t* test. (**B**) Spearman’s correlation test between LGR5 and RSPO3 protein expressions in the Malmö-CC cohort. (**C**) Spearman’s correlation test between LGR5 and RSPO3 mRNA expressions using the publicly available GSE44076 cohort. (**D**) Spearman’s correlation test between RSPO3 and WNT5A mRNA expressions using the publicly available GSE44076 cohort. (**E**) RSPO3 mRNA expression in HT-29 colon cancer cells stimulated in vitro with or without 100 µM Foxy5 for 24 h. The result in the graph is shown as the mean ± SEM for at least three independent experiments. (**F**) RSPO3 mRNA levels in HT-29 colon cancer tissues from xenograft mice treated with vehicle or Foxy5 (2 µg/g) every other day for a 14-day period. The results in the graph are shown as the mean ± SEM. (**G**) Representative IHC images of RSPO3 protein expression in the HT-29 cell xenografts from mice treated with vehicle or Foxy5 (2 µg/g) every other day for a 14-day period. The results in Panels (**F**,**G**) are presented in the graph as the mean ± SEM, and 5 xenograft tissues from 5 different mice were analyzed per group of animals. * *p* < 0.05, ** *p* < 0.01, analyzed with Mann-Whitney unpaired Student’s *t* test.

**Figure 6 cells-12-02658-f006:**
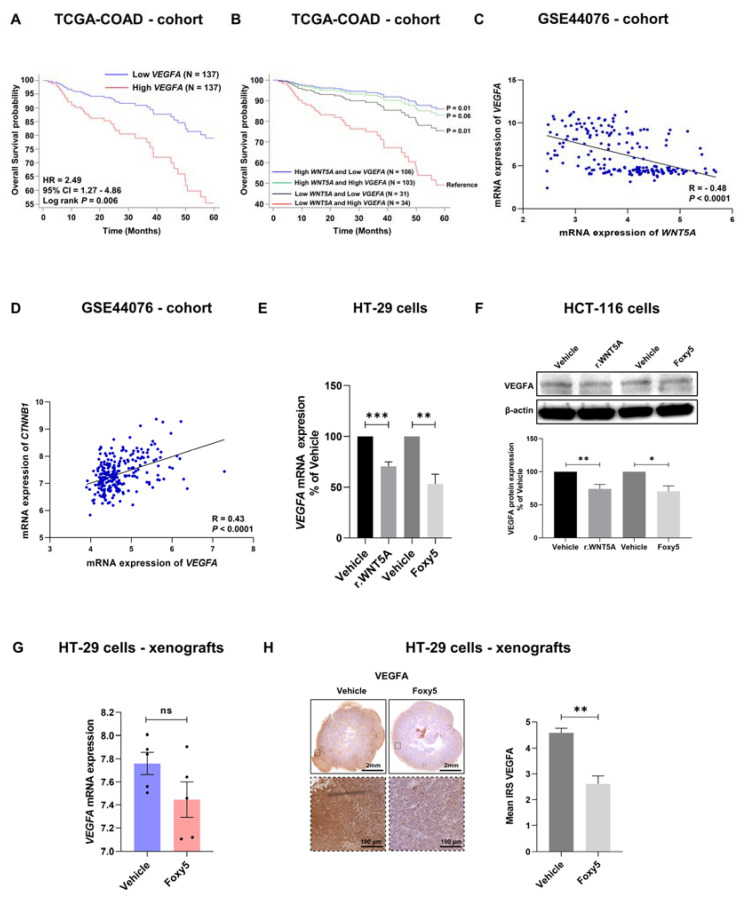
The expression of the β-catenin target VEGFA predicts a poor prognosis, correlates with the expression of WNT5A and β-catenin at the transcript level in colon cancer tissue, and is downregulated by WNT5A signaling. (**A**) Using the TCGA-COAD cohort, we performed univariate five-year overall survival analysis for VEGFA. (**B**) Five-year overall survival analysis of 4 possible patient groups from the TCGA-COAD cohort based on the combinations of VEGFA (low and high) and WNT5A (low and high) mRNA expressions. Spearman’s correlation tests are shown between (**C**) VEGFA and WNT5A mRNA expression levels and (**D**) VEGFA and β-catenin mRNA (CTNNB1). Both correlations were made based on data from the publicly available GSE44076 cohort. (**E**) This panel outlines VEGFA mRNA expression in HT-29 colon cancer cells stimulated in vitro with or without 400 ng/mL rWNT5A or 100 µM Foxy5 for 24 h. (**F**) Representative Western blot and the accumulated densitometric VEGFA protein expression in HCT-116 colon cancer cells stimulated in vitro with or without 400 ng/mL rWNT5A or 100 µM Foxy5 for 24 h. The results in the graphs are shown as the mean ± SEM for at least three independent experiments. (**G**) VEGFA mRNA levels in HT-29 colon cancer tissues from xenograft mice treated with vehicle or Foxy5 (2 µg/g) every other day for a 14-day period. (**H**) Representative IHC images of VEGFA protein expression in the HT-29 colon cancer cell xenografts from mice treated with vehicle or Foxy5 (2 µg/g) every other day for a 14-day period. The results in panels (**G**,**H**) are presented in the graphs as the mean ± SEM (5 xenograft tissues from 5 different mice were analyzed per group of animals); * *p* < 0.05, ** *p* < 0.01 and *** *p* < 0.001, analyzed with Mann-Whitney unpaired Student’s *t* test.

**Figure 7 cells-12-02658-f007:**
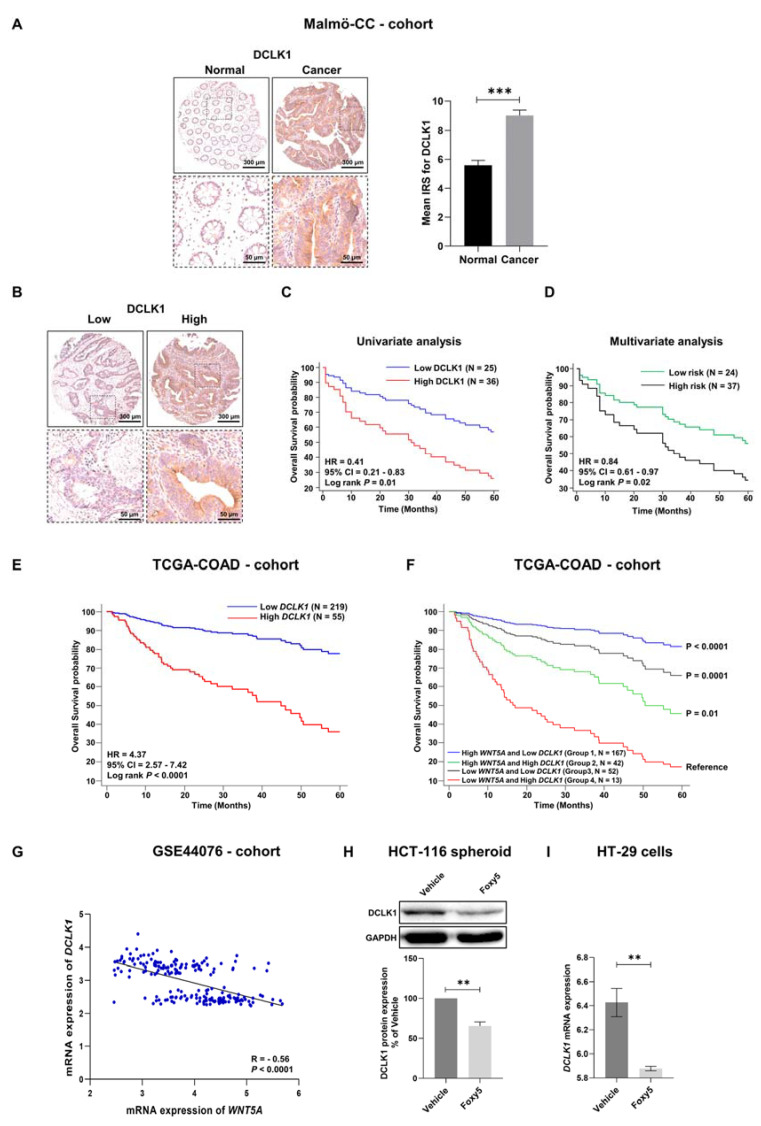
The expression of the colon cancer stem cell marker DCLK1 predicts a poor prognosis, correlates with the expression of WNT5A in colon cancer tissue, and is downregulated by Foxy5. (**A**) Representative IHC staining images of the DCLK1 protein in normal mucosa and matched colon cancer tissue and their mean IRS values are presented in the graph. The results are given as the mean ± SEM; *** *p* < 0.001, analyzed with Wilcoxon match-paired Student’s *t* test. (**B**) Representative IHC staining images of low and high DCLK1 protein expression in colon cancer tissue. (**C**) Univariate five-year overall survival analysis of patients with either low or high DCLK1 expression in their tumors. (**D**) Multivariate five-year overall survival analysis for low-risk (low DCLK1 expression) and high-risk (high DCLK1 expression) patients when adjusted for sex, LNM and TNM stage (Appendix A). (**E**) Using the TCGA-COAD cohort we performed univariate five-year overall survival analysis for DCLK1. (**F**) Five-year overall survival analysis of four possible groups from the TCGA-COAD cohort based on the combinations of DCLK1 (low and high) and WNT5A (low and high) mRNA expressions. (**G**) Spearman’s correlation test between DCLK1 and WNT5A mRNA expressions based on data from the publicly available GSE44076 cohort. (**H**) HCT-116 colon cancer cells were grown to form colonospheres, and then the cells were treated with either vehicle or 100 µM Foxy5. (**I**) DCLK1 mRNA expression in HT-29 colon cancer cells stimulated in vitro with or without 100 µM Foxy5. The effects of Foxy5 treatment on DCLK1 expression were analyzed by Western blotting (panel **H**) or RT-qPCR (panel **I**) and the results are presented in the graphs as the mean ± SEM for at least three independent experiments; ** *p* < 0.01, analyzed with Mann-Whitney unpaired Student’s *t* test.

## Data Availability

The datasets used and/or analyzed during the current study are available within the article or Appendix A or from the corresponding author upon reasonable request.

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
