# Peer review of "LGR5 Expression Predicting Poor Prognosis Is Negatively Correlated with WNT5A in Colon Cancer"

_cells, 2023, doi:10.3390/cells12222658_

Round 1

Reviewer 1 Report

Comments and Suggestions for Authors

In this study, the authors explored the mechanisms underlying the crosstalk between Wnt5A (and its hexapeptide mimetic Foxy5) and the Wnt/β-catenin signaling pathway, focusing primarily on the G protein-coupled receptor LGR5 and its ligand RSPO3. The data provide strong evidence for the impact of Wnt5A on the expression of LGR5 and its ligand RSPO3. In particular, immunocytochemistry and RT-qPCR studies performed on normal and colon cancer human samples, as well as multivariate analyses based on risk score, provide significant evidence of the prognostic value of Wnt5A and LGR5 expression. Taken together, the data indicate that Wnt5a signaling impacts colon cancer stemness, which is of both fundamental and clinical interest, given that the Wnt5A-mimicking peptide Foxy5 is currently undergoing phase II clinical trials.

My main concern lies in the fact that 1) in vitro and in vivo data were systematically obtained using a single cell line and 2) different cell lines were used for each experimental procedure, which raises the question of the consistency of the results: clonogenicity, colonosphere formation and qRT-PCR were performed only with cultured HCT116, xenografts were done only with HT29 cells and Immunocytochemistry was carried on only with cultured SW480 cells. For consistency, I would therefore recommend to repeat/validate all in vitro data (clonogenicity, colonosphere formation and immunocytochemistry) with cultured HT29 cells.

In the second part of the article, the authors explore the link between Wnt5A signaling and the expression of the vascular endothelial growth factor A (VEGFA) and the colon cancer stem cell marker DCLK1. Although the VEGFA data are also of clinical interest, they appear to be outside the scope of the article, which focuses primarily on LGR5/RSPO3. Moreover, results about VEGFA are neither included and discussed in the discussion section nor mentioned in the title of the article.

Major points:

§ 3. Results

Q2- § 3.2. The authors conclude that Wnt5A regulates expression of the cell surface receptor LGR5 (lines 345-346). However, can the authors give a possible explanation about why some patients with high levels of wnt5A also have high levels of LGR5? Conversely, why do some patients with low levels of wnt5A have high levels of LGR5?

Q3- § 3.3.  In vitro and in vivo studies provide strong evidence that activation of Wnt5A signaling (either using recombinant Wnt5A or the Foxy5 peptide) decreases LGRF5 expression and has anti-tumor activity. However, as mentioned above, I don't quite understand why the authors only used HCTT16 cells in their in vitro assays whereas they xenografted HT29 cells for their in vivo experiments. To ensure consistency between in vitro and in vivo experimental data, I would recommend repeating/validating the clonogenicity and colonosphere formation studies with HT29 cells.

Q4- § 3.4. Again, and as mentioned above, it is surprising that the authors still used an additional cell line, namely SW480, to study Foxy5-induced effects on LGR5 expression and β-catenin signaling. For consistency of experimental data, I would recommend repeating immunocytochemistry with at least HT29 cells (and HCT116 cells, if possible).

Q5- § 3.3 and 3.4. Could the authors provide additional data regarding RSP03 expression from their in vitro and in vivo assays (clonogenicity, colonosphere formation, immunocytochemistry, xenograft tumor growth studies)?

Q6- § 3.6. Lines 453 to 455: in reference to figure 6E, the authors state that " they stimulated HT29 with recombinant Wnt5A or Foxy5". However, Figure 6E shows data with HCT116 cells instead of HT29 cells. Could the authors correct this point and further provide additional data with HT29 cells in order to validate the result and present in vitro data consistent with in vivo data (xenografts performed with HT29 cells)?

Minor point:

Table S1: please define HR and CI in the legend.

Reviewer 2 Report

Comments and Suggestions for Authors

The study presented by Mehdawi and colleagues explores the role of WNT/β-catenin signaling in colon cancer development and progression. WNT5A, a potential tumor suppressor, and its mimic Foxy5 were investigated for their effects on LGR5 and RSPO3, two promoters of β-catenin signaling. The researchers found that WNT5A signaling negatively correlates with the expression of LGR5 and RSPO3 in colon cancer patients. In experiments, WNT5A signaling suppressed β-catenin activity, LGR5, RSPO3, and VEGFA expression, as well as the formation of colonies and spheroids. Since β-catenin signaling is linked to cancer stemness, the study also explored the relationship between WNT5A and the cancer stem cell marker DCLK1. They found that DCLK1 expression was negatively correlated with WNT5A expression and was experimentally reduced by WNT5A signaling. The results suggest that WNT5A and Foxy5 may have potential as treatment strategies for colon cancer patients by reducing LGR5/RSPO3 expression, inhibiting stemness, and suppressing VEGFA expression.

The data presented here are novel and convincingly enhance our understanding of the mechanism of action of Foxy5. The manuscript is very well-prepared, and I have neither criticisms nor additional comments.

Reviewer 3 Report

Comments and Suggestions for Authors

Lubna M. Mehdavi et al. presented a manuscript entitled "LGR5 expression predicting poor prognosis negatively correlates with WNT5A expression in colon cancer tissue: the WNT5A signaling pathway reduces LGR5 expression", dedicated to the study of the possible mechanism of participation of the WNT5A ligand of the non-canonical Wnt signaling pathway in the activation or inhibition of canonical Wnt/β-catenin signaling for colon cancer.

In general, the manuscript is well written, has a good scientific soundness, delivers interesting outcomes and deserves to be published after the revisions. There are some important points that need consideration.

  1. Titul. I would suggest shortening the name. For example, "LGR5 expression predicting poor prognosis in colon cancer is negatively correlated with WNT5A"

  1. Abstract. It must be emphasized that the effects of WNT5 (ligand of non-canonic Wnt signaling) and its mimetic Foxy5 were studied not only on such Wnt/β-catenin signaling pathway molecules as the activator LGR5 receptor and its ligand RSPO3, but also β-catenin and its target gene VEGFA.

Line 14. Clarify which cohorts of patients with colon cancer were analyzed.

Line 15. Clarify that 2 colon cancer cell lines were used for in vitro experiments and 1 colon cancer cell line – for in vivo ones.

Check whether gene names are written in italics

  1. Keywords. «β-catenin signaling» should be changed to «Wnt/β-catenin signaling» and Foxy5 should also be added

  1. Section “Introduction”.

Line 60-71. «…the fact that WNT/β-catenin-independent signaling has been documented to cross-talk with and inhibit the WNT/β-catenin-dependent signaling pathway [16].»

In my opinion, there is no indication of this crosstalk in reference [16]. Please provide adequate links, for example,

https://pubmed.ncbi.nlm.nih.gov/27575935/

https://pubmed.ncbi.nlm.nih.gov/24896565/

https://www.ncbi.nlm.nih.gov/pmc/articles/PMC5752873/

https://pubmed.ncbi.nlm.nih.gov/29113510/

It could be mentioned that the LGR5 is a negative regulator of tumourigenicity, antagonizes Wnt signalling and regulates cell adhesion in colorectal cancer cell lines

https://pubmed.ncbi.nlm.nih.gov/21829496/

Lines 90-94. . This is part of the "Introduction" section needs improvement. The definition of the objectives of this document should be separated from the conclusions reached as a result of research.

  1. Section “Materials and Methods”.

Line 123. Please provide information on where human HT-29 colon cancer cells were purchased.

  1. Section “Results”

Line 350. "To test the above hypothesis".

Please write in more detail what hypothesis you tested.

In Section 3.4, when describing immunofluorescence images in Figure 4A, it would be worthwhile to indicate whether Foxy5 affects cell morphology and discuss this aspect. This is difficult to understand, since there are no images of cells in a bright field to control cell morphology.

Line 468. “Figure 5. The expression of the LGR5 ligand RSPO3 correlates with the expression of LGR5 and WNT5A ……”

In the title of Figure 5, it is worth pointing out that the expression of the RSPO3 correlates positively with the expression of LGR5 and negatively with the expression of WNT5A.

7. In the "Discussion" section, unforgivably little attention is paid to the VEGF A. It is worth eliminating this drawback. Lines 465-466 would be appropriate here.

Round 2

Reviewer 1 Report

Comments and Suggestions for Authors

Title: LGR5 expression predicting a poor prognosis is negatively correlated
with WNT5A expression in colon cancer tissue: WNT5A signaling reduces LGR5
expression
Authors: Lubna M Mehdawi *, Souvik Ghatak, Payel Chakraborty, Anita
Sjölander, Tommy Andersson *

is acceptable in its present/corrected version.